# Topological to magnetically ordered quantum phase transition in antiferromagnetic spin ladders with long-range interactions

**Luhang Yang**[⋆†]**, Phillip Weinberg**[†] **and Adrian E. Feiguin**

Department of Physics, Northeastern University,
Boston, Massachusetts 02115, USA

⋆ yang.luh@northeastern.edu    †: Authors contributed equally.

## Abstract

We study a generalized quantum spin ladder with staggered long range interactions that decay as a power-law with exponent $\alpha$. Using large scale quantum Monte Carlo (QMC) and density matrix renormalization group (DMRG) simulations, we show that this model undergoes a transition from a rung-dimer phase characterized by a non-local string order parameter, to a symmetry broken Néel phase. We find evidence that the transition is second order. In the magnetically ordered phase, the spectrum exhibits gapless modes, while excitations in the gapped phase are well described in terms of triplons – bound states of spinons across the legs. We obtain the momentum resolved spin dynamic structure factor numerically and find a well defined triplon band that evolves into a gapless magnon dispersion across the transition. We further discuss the possibility of deconfined criticality in this model.

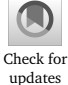

# 1   Introduction

The study of exotic phases of matter of quantum origin is one of the cornerstones of modern condensed matter physics, motivating a quest for materials and models that could exhibit novel unconventional properties, such as fractionalized excitations that cannot be described as Landau quasiparticles, topological states that do not admit a local order parameter, and quantum phase transitions that defy the Landau-Ginzburg paradigm.

Quantum magnets exhibit a vast and varied phenomenology and offer a relatively simple and intuitive playground where to test and verify these ideas. A prototypical example of phase transition that has been extensively studied is the one between a disordered dimer phase and a Néel ordered antiferromagnet(AFM) [1–3]. On both sides of the critical point, excitations carry spin $S = 1$: triplons in the magnetically disordered gapped phase; gapless magnons in the ordered phase. At the transition, besides two gapless Goldstone modes, a massive amplitude mode (also referred-to as the Higgs mode) is expected. Remarkably, this behavior has has been experimentally observed under pressure in TlCuCl$_3$ [4–7].

A crucial reason explaining why the theoretical study of these phenomena has been limited to two and three spatial dimensions is justified by the Mermin-Wagner theorem [8], that establishes that quantum Hamiltonians with short range interactions cannot spontaneously break a continuous symmetry in dimensions lower than $D = 2$. Even in 2D systems, this can only occur at zero temperature $T = 0$. In this work, we circumvent these restrictions by introducing long range non-frustrating interactions to the problem. We can thus conceive a ladder Hamiltonian that exhibits true long range Néel order and apply numerical techniques that are well suited for studying low-dimensional problems. Explicitly, the model of interest is a conventional Heisenberg ladder with additional algebraically decaying all-to-all couplings:

$$H = -J \sum_{i>j} \frac{(-1)^{|x_i+y_i-x_j-y_j|}}{|\vec{r}_i - \vec{r}_j|^\alpha} \vec{S}_i \cdot \vec{S}_j, \tag{1}$$

where the spin operators $\vec{S}_i$ are localized at positions $\vec{r}_i = (x_i, y_i)$ on a two leg $2 \times L$ ladder with $y_i = 1, 2$. The alternating sign on the interactions ensures that they will be AFM between spins on opposite sublattices, and ferromagnetic otherwise (See Fig. 1 for a graphical representation). One could in principle envision such interactions emerging from a proximity coupling with a higher dimensional antiferromagnet or other ladders in a perturbative sense. The only free parameter in the problem is the exponent $\alpha$; for large $\alpha$ we expect the ground state to be in the same phase as the conventional Heisenberg ladder and the physics is well understood: the correlation length is short, of a few lattice spaces, and the gap is of the order of the coupling $J$ [9–20]. This ground state is adiabatically connected to the trivial limit of the conventional Heisenberg ladder corresponding to anisotropic couplings along the legs and rungs $J_{rung} \gg J_{leg}$. In this "strong rung coupling limit" the ground state is a product of rung dimers, the single-triplet gap is of order $\mathcal{O}(J_{rung})$ and excitations are rung triplets that can propagate coherently along the ladder.

Notoriously, unlike the case of dimerized chains, this "rung singlet phase" does not break any lattice symmetry, and even though it is adiabatically connected to a product state in the limit of $J_{rung} \to \infty$, it is characterized by a broken "hidden" symmetry [21–25] described in terms of a "string" order parameter [26–28]:

$$O = \lim_{l \to \infty} O_l, \qquad O_l = -\left\langle \tilde{S}_0^z \left( \prod_{j=1}^{l-1} e^{i\pi \tilde{S}_j^z} \right) \tilde{S}_l^z \right\rangle, \tag{2}$$

with $\tilde{S}_j^z = S_{j+1,1}^z + S_{j,2}^z$ connects spins along one of the diagonals between two rungs. The connection between ladders and the topological aspects of the Haldane chain were noticed while back [26].

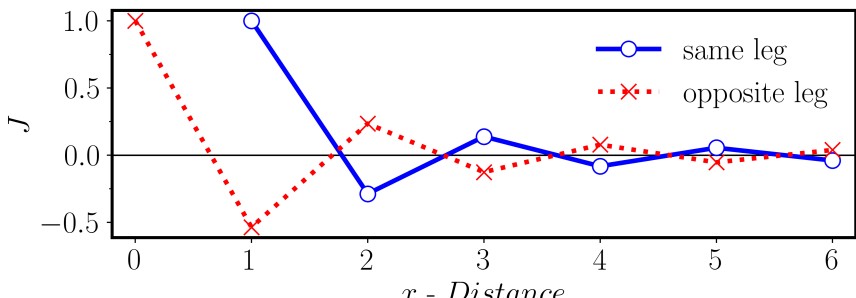

Figure 1: Exchange interaction between two spins at distance $x$ along the same and opposite leg, for a value of $\alpha = 1.8$.

On the other hand, in the model described by Eq.(1) we anticipate that the all-to-all unfrustrating interactions will yield a ground state with long-range AFM (Néel) order and gapless excitations for relatively small $\alpha$. These expectations are based on previous studies of Hamiltonian (1) in 1D chains [29–34], where a transition between a gapless spin-liquid and a gapless ordered phase was revealed.

In this work, we focus on identifying and characterizing the quantum critical point, as well as understanding the excitation spectrum at and away from the transition. Given that the model described by Eq.(1) does not have a sign problem we use QMC to study the properties of the transition while using the time-dependent density matrix renormalization group method (tDMRG) [35–38] to understand the low-energy excitations. The behavior of the gap and order parameters is discussed in Sec. 2, offering compelling evidence for a continuous quantum phase transition between the Néel and rung-dimer phases at $\alpha_c = 2.519(1)$. In Sec. 4 we present results for the dynamic spin structure factor $S^z(q, \omega)$. We finally close with a summary and discussion of our findings.

## 2 Quantum critical point

In this section, we present several complementary methods to estimate the position of the quantum critical point $\alpha_c$. We first focus on determining the critical point using the correlation length exponent $\nu$ using QMC on periodic ladders of length up to $L = 96$. We also develop a method to determine the critical point using the dynamic exponent calculated from finite size gaps obtained from DMRG [39, 40] calculations.

To study the ground state properties we use projector QMC with a trial state that is given by an amplitude product state in the valance bond basis [32, 41, 42]. We use the same trial state for all values of $\alpha$. This state has long range Néel order to help reduce the number of projector steps needed to reach the ground state in the Néel phase. In the rung-dimer phase, this trial state has little effect on the number of projector steps required because of the finite gap in the thermodynamic limit. We study two different order parameters that define the order on either side of the transition. If the transition point is determined to be the same using both order parameters, we can exclude the possibility of an intermediate phase between the rung-dimer and Néel phases. For the Néel order we use the Binder Cumulant defined as:

$$B = \frac{3}{2} \left( 1 - \frac{1}{3} \frac{\langle M_s^2 \rangle}{\langle |M_s| \rangle^2} \right), \tag{3}$$

where $M_s = \sum_i (-1)^{x_i + y_i} S_i^z$. In our QMC simulations we exploit the full $SU(2)$ symmetry of the ground state [41, 42].

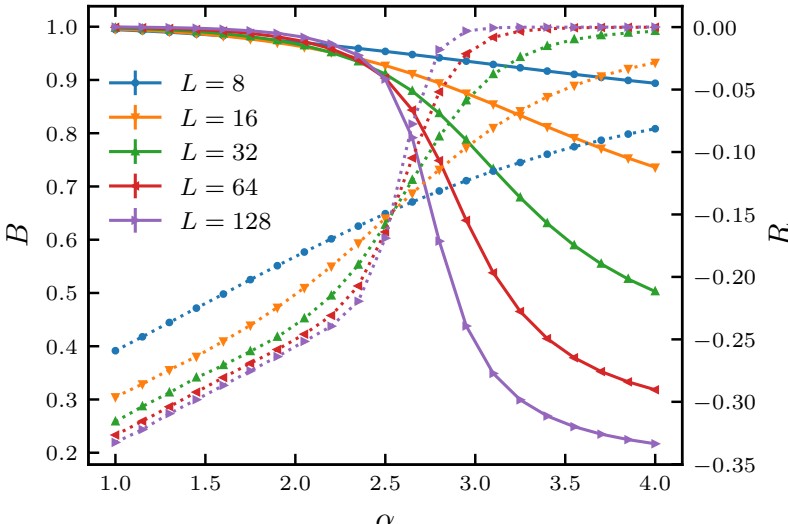

Figure 2: Binder Cumulant $B$ (solid lines) and the log string-order parameter ratio $R$ (dotted lines) as a function of $\alpha$ for different system sizes. Note the scale for $B$ and $R$ are on the left and right side of the frame respectively. The error bars for both quantities are smaller than the symbols.

For the rung-dimer phase we use the string order parameter $O$, defined in Eq.(2). Much like a correlation function, we observe that in a finite size system $O_l$ has a non-vanishing value due to finite-size effects. To systematically study the convergence to the thermodynamic we look at the ratio between $O_l$ measured at two lengths, $L/2$ and $L/4$, and analyze the following quantity as a function of $\alpha$ and system size $L$ [43]:

$$R = \log\left(\frac{O_{L/2}}{O_{L/4}}\right). \tag{4}$$

In order to measure $O_l$ we use the standard estimator calculated from the $S^z$ basis of the QMC simulation. Because of the non-local nature of this order parameter, the results have more noise compared to the Binder cumulant.

Considering a functional form $O_l = f_l + C$ (where $f_l$ is an asymptotically decaying function) there are a few possible outcomes for $R$ in the thermodynamic limit. For $C > 0$, $R \to 0$ as $L \to \infty$. In the other hand, for $C = 0$ the asymptotic behavior of $R$ is determined by the behavior of $f_l$. If $f_l$ decays as a stretched exponential (or faster) $R$ diverges as $L \to \infty$, while for $f_l$ decaying slower, $R \to$ const $\leq 0$, *i.e.* for a power-law decay this constant will be negative.

In Fig. 2 we show both $B$ and $R$ as a function of $\alpha$ for various system sizes. We observe that $B$ monotonically increases for decreasing $\alpha$. This behavior indicates the onset of long-range Néel order for small values of $\alpha$. On the other hand, $R$, which is always negative by definition, is growing in absolute value as $\alpha$ decreases. For larger values of $\alpha$, in the gapped phase, $R$ tends to 0 with increasing system size, implying that $O > 0$. In the Néel phase (small $\alpha$) $R$ converges to a finite value, indicating that $O = 0$. The "steepness" of the $B$ and $R$ curves increases with increasing system size. This observation is consistent with the finite-size behavior one would expect from a phase transition [43].

Both $B$ and $R$ represent different types of order and they serve as means to independently determine the critical point. If the transition occurs between two ordered phases, one would expect that $B$ and $R$ will share the same critical point and correlation length exponent $\nu$. We have specifically chosen the forms for $B$ and $R$ to allow us to systematically extract the critical point and correlation length exponent from finite-size calculations.

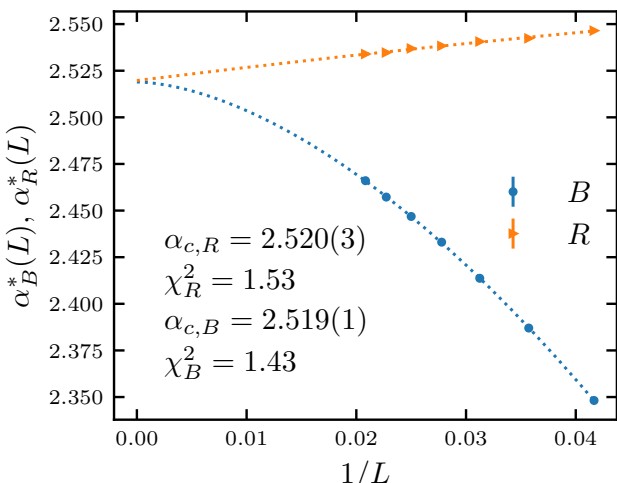

Figure 3: Crossing points for both the Binder cumulant $B$ (circles) and the log string-order parameter ratio $R$ (triangles) as a function of $1/L$ for different system sizes. The dashed lines show the extrapolation to the thermodynamic limit and the legend indicates the extrapolated value along with the normalized $\chi^2$ value for the fit.

To determine the location of the critical point we look at the crossing points between two curves (either $B$ or $R$) corresponding to different system sizes. Because of finite-size scaling (FSS) corrections, the crossing points will drift closer to the critical point as the system sizes increase [43]. Specifically, we look at the crossing points between curves corresponding to system sizes $L$ and $2L$, that we denote as $\alpha^*(L)$.

We can also extract the correlation length exponent $\nu$ from $B$ and $R$ because they both have a scaling dimension of 0. From the FSS form for this type of observable it is possible to show that the value of $\nu$ can be determined by the following limit:

$$\nu = \lim_{L \to \infty} \nu^*(L), \qquad \nu^*(L) = \left[ \log_2 \left( \frac{\partial_\alpha Y(\alpha^*(L), 2L)}{\partial_\alpha Y(\alpha^*(L), L)} \right) \right]^{-1}, \tag{5}$$

where $Y(\alpha, L)$ corresponds to either $B$ or $R$. This can easily be seen by looking at the FSS form for an observable with scaling dimension 0 [43].

Since in practice one can only study a finite number of values of $\alpha$, we interpolate those points with a polynomial. Using the interpolation for $L$ and $2L$ we can calculate both $\alpha^*(L)$ and $\nu^*(L)$. To account for the statistical errors coming from the QMC sampling we use the bootstrapping method. This involves drawing a new set of values for $B$ and $R$ from a normal distribution with a mean and standard deviation given by the QMC mean and standard error for each point respectively. After drawing the new points, a polynomial is fitted from which $\alpha^*(L)$ and $\nu^*(L)$ are obtained. This procedure is repeated for many random realizations of the data points. Each realization has independent values of $\alpha^*(L)$ and $\nu^*(L)$. From this set of values the mean and standard deviation are calculated. In this work we use 10000 realizations to generate the mean and standard deviation corresponding to the points and error bars shown in Figs. 3 and 4.

The results for $\alpha^*(L)$ and their respective estimates in the thermodynamic limit for both $B$ and $R$ are show in Fig. 3. The extrapolation is done using a power-law fit of the form $\alpha^*(L) = \alpha_c + b/L^\gamma$ for both, $B$ and $R$. The two order parameters give a critical point of 2.520 within error bars. We also use a linear expression to extrapolate the values of $\nu^*(L)$ for both $B$ and $R$, as shown in Fig. 4, yielding a value of $\nu = 1.79$ within error bars. These results are a strong indication that there is a direct transition between the rung-dimer phase and the Néel

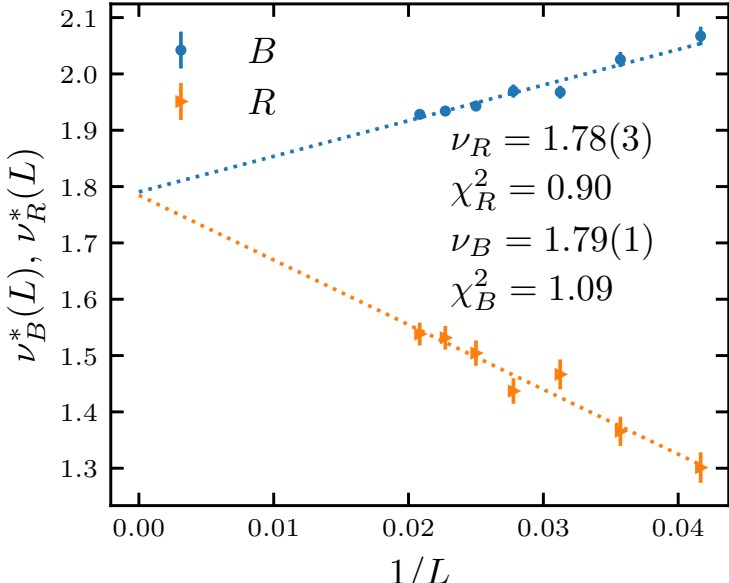

Figure 4: Extrapolation of the exponent $\nu^*(L)$ for both the Binder cumulant $B$ (circles) and the log string-order parameter ratio $R$ (triangles) as a function of $1/L$ for different system sizes. The legend indicates the extrapolated values for $\nu$ along with the normalized $\chi^2$ values for each fit.

phase. Our next goal is to establish if the transition is continuous or first order.

## 3 Gap and dynamic exponent $z$

The dynamic exponent $z$ can provide useful information about the behavior of excitations as well as whether or not the transition is continuous. In order to obtain $z$ we use finite size extrapolations of the spin-triplet gap, calculated using the DMRG method. What makes this problem particularly challenging is the possibility of a volume law entanglement law due to the presence of all-to-all interactions. However, in the gapped phase, the correlation length remains finite and the entanglement remains under control. Surprisingly, the entanglement entropy does not grow dramatically in the gapless phase and across the transition. This may appear to be a general feature of one-dimensional models with long-range interactions as has been observed in quantum spin chains, which display a $log(L)$ behavior [33,44–46]. The main numerical cost lies on the fact that the number of terms in the Hamiltonian grows quadratically with system size. In the calculations presented here we have studied ladders of size $L \times 2$ sites with $L$ up to 48, with open boundary conditions and adjusting the bond dimension such that the truncation error is kept under $10^{-7}$, translating into up to 1000 states.

As discussed in the previous subsection, in the limit $\alpha \to \infty$ the problem reduces to the conventional Heisenberg ladder Hamiltonian with nearest neighbor interactions. As the value of $\alpha$ is decreased, the antiferromagnetic correlations are enhanced and the gap is reduced. In the left panel of Fig. 5 we show the behavior of both the gap extrapolated to the thermodynamic limit as a function of $\alpha$. To carry out the extrapolation to the thermodynamic limit we use a second order polynomial fit of finite-size data as a function of $1/L$, as shown in the inset. We do not see a closing of the gap at $\alpha = 2.520$ due to the strong sublinear scaling behavior that introduces corrections that require larger system sizes (as we describe below, a power-

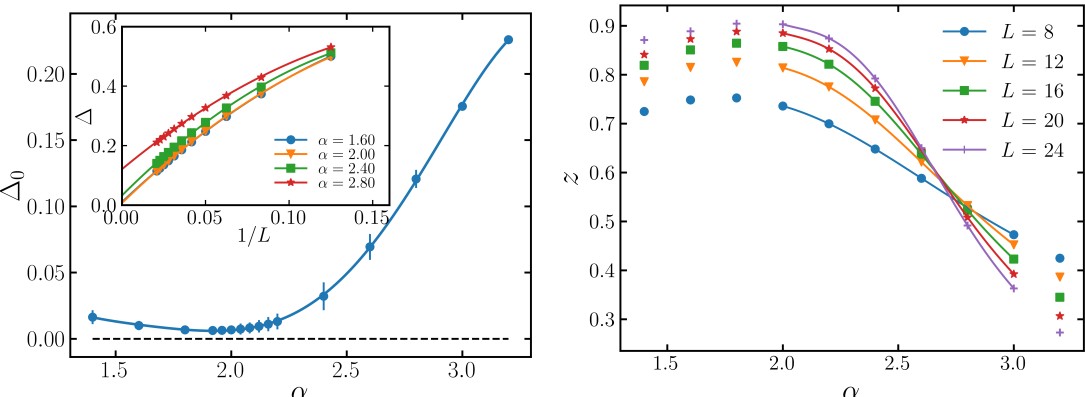

Figure 5: (left) Gap extrapolated to the thermodynamic limit as a function of $\alpha$. The inset shows the finite-size gaps, obtained with DMRG, used in the extrapolation of $\Delta_0$ with second order polynomial fit. (right) Dynamic critical exponent $z(L)$ as a function of $\alpha$ for various system sizes. The circles correspond to $z(L)$ calculated using the gaps obtained from DMRG. The curves are the best fit using fourth order polynomials.

law extrapolation is ill-conditioned for this small dataset). We point out that the upturn of the curve for small $\alpha$ is likely an artifact of the extrapolation that becomes less reliable as the spectrum becomes more singular at the ordering wave vector. Even though, once the system orders, the spectrum is expected to remain gapless, we do not discard the possibility of a gap reopening for small $1 < \alpha < 2$, since the long range interactions violate Goldstone's theorem hypotheses and symmetry breaking could be accompanied by a gap [47–49] (we discuss this point in more detail in the Conclusions).

Given the power-law nature of the interactions, the finite-size effects are much stronger compared to a local Hamiltonian making the dynamic exponent difficult to extract. We can account for these corrections by expressing the gap as:

$$\Delta(L) = a L^{-z}(1 + f_\Delta(L)). \tag{6}$$

Here we include all finite-size corrections in $f_\Delta(L)$ such that, in the thermodynamic limit, $f_\Delta(L) \to 0$. Instead of fitting the gap directly, we can define an approximation of the dynamic exponent for a finite-size system by calculating the log of the ratio of the gap between system sizes $L$ and $2L$,

$$z(L) \equiv \log_2\left(\frac{\Delta(L)}{\Delta(2L)}\right) = z + \log_2\left(\frac{1 + f_\Delta(L)}{1 + f_\Delta(2L)}\right). \tag{7}$$

When $L \to \infty$ the second term on the right side will vanish. Using $z(L)$ allows one to directly extrapolate the dynamic exponent removing any bias in trying to guess the functional form of finite-size corrections.

At a transition between a gapless and gapped phase, $z$ will have a discontinuous jump at the critical point from 0 to a finite value, much like the Binder cumulant for an order parameter. For finite-size systems the non-analytic behavior becomes smooth but the evidence of this discontinuity becomes more pronounced as system sizes become larger. As a result, we use the crossing points between two system sizes to extrapolate the location of the critical point. The crossing points in the right panel of Fig. 5 indicate that the dynamic exponent is going to 0 above $\alpha_c$, while approaching a value larger than 0 below the transition.

We can also use $z(L)$ to determine whether this transition is first or second order. In the continuous case, $z(L)$ at the critical point will tend towards a finite value in the thermodynamic

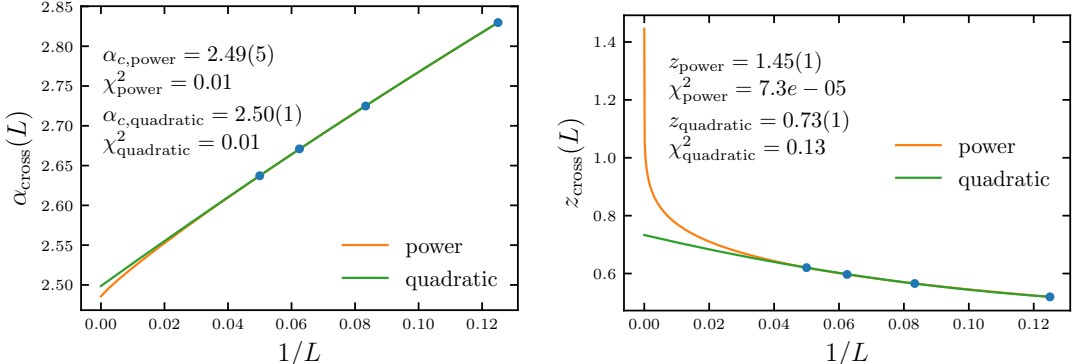

Figure 6: (left) Extrapolation of crossing point of the curves in the right panel of Fig. 5 to the thermodynamic limit using power-law (e.g. $\alpha_{\text{cross}}(L) = \alpha_c + b/L^{\beta}$) and quadratic fits shown in the plot as orange and green lines respective. (right) Extrapolation of $z(L)$ values at the crossing points with a power-law and quadratic fits shown in the plot as orange and green lines respective. The error bars for each point are calculated from the co-variance matrix of the polynomial interpolations and the error bar for the critical point and dynamic exponent in the figure have been calculated the bootstrap method. The $\chi^2$ value for the fit is shown in the figures. We have normalized it by the number of left-over degree's of freedom, in this case is one.

limit while for a first order transition $z(L)$ will tend to infinity. There is no indication in our results of a divergence in the dynamic exponent for any of the values of $\alpha$ we looked at, thus providing evidence for a second order transition.

Using the crossing points between system sizes $L$ and $L + 4$, we can estimate the critical point by extrapolating them to the thermodynamic limit as a function of $1/L$. We show the results for our extrapolation in the left panel of Fig. 6. To fit the data we use an expression $y = y_0 + b/L^{\beta}$ due to the limited number of points. We find that the critical point is around 2.5 based on the two different extrapolation methods.

It is worth noting that a value of $z = 1$ would indicate the possibility of an underlying conformal invariance and, consequently, a deconfined quantum critical point. To determine the value of $z$ at the critical point one can extrapolate the values of $z(L)$ at the crossing points just as we did for the Binder cumulant [43]. The results are shown in the left panel of Fig. 6. Unlike in the critical point estimate, the extrapolated values differ significantly between the power-law and the quadratic extrapolations indicating there are larger finite-size corrections to this quantity. In this case, we cannot provide an accurate estimate for the critical exponent.

## 4 Spin dynamics

In order to calculate the spin dynamic structure factor we used the time-dependent DMRG method (tDMRG) [35, 36], a well established technique described in detail in the original work Refs. [35, 50] and reviews Refs. [37, 38]. The longitudinal two-time spin-spin correlation function is defined as:

$$\langle S_r^z(t)S_0^z(0)\rangle = \langle\psi_0|e^{iHt}S_r^z e^{-iHt}S_0^z|\psi_0\rangle, \tag{8}$$

where we take $S_0^z$ at the center of the one of the legs of the ladder, and $r$ is the distance from the middle. The spectral function is obtained by Fourier transforming from real space and time to momentum space and frequency. This procedure is carried out over a finite time range (in

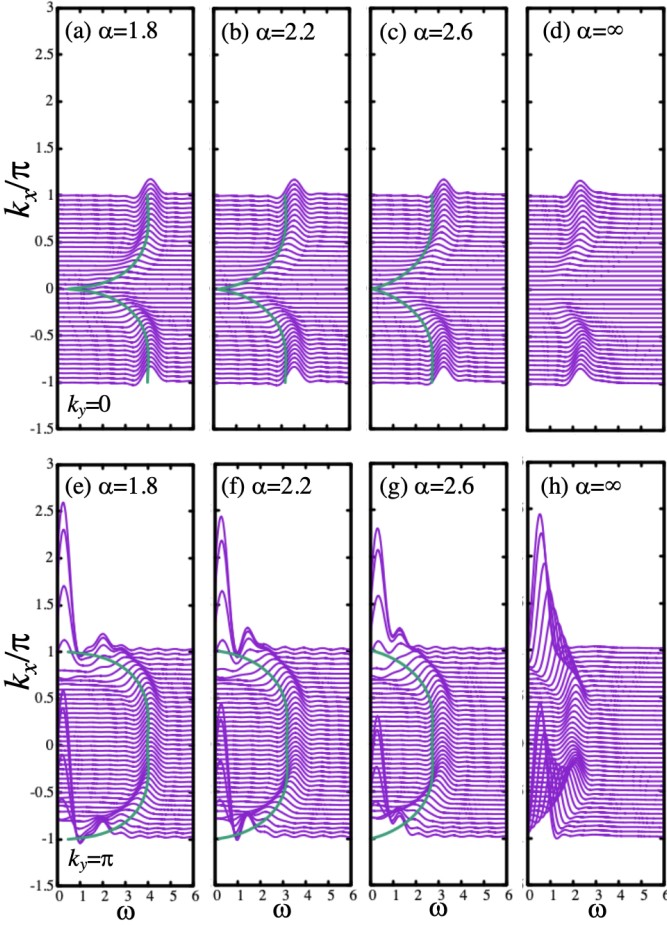

Figure 7: Longitudinal dynamic spin structure factor $S(k, \omega)$ for a $20 \times 2$ ladder with long range interactions and different exponent $\alpha$ across the quantum critical point, obtained with tDMRG. Upper(lower) row show the symmetric (antisymmetric) channel. Ringing at high energies is due to the finite time integration window (see text). Also shown are the linear spin-wave dispersion and results for the conventional ladder with only nearest-neighbor terms, $\alpha = \infty$.

our case $t_{max} = 10$). For this reason, the poles in the spectral function will not be well defined deltas, but will display artifacts such as artificial ringing that can be attenuated by means of standard windowing techniques also used in signal processing. As a consequence, the width of the spectral features will be inversely proportional to the width of our time window. Due to the long-range nature of the terms in the Hamiltonian, we employ a time-step targeting procedure with a Krylov expansion of the time-evolution operator [50]. We fixed the time step $\delta t = 0.05$ (measured in units of $J^{-1}$). We fixed the maximum truncation error to $10^{-7}$ in the time range considered. All results shown here are for a relatively small ladder of length $L = 20$. Unlike the ground-state calculations, the entanglement entropy grows rapidly in time, together with the number of states required to keep the error under bounds which can be as large as $m = 1500$. In addition, as mentioned before, the number of terms in the Hamiltonian makes the time evolution very time consuming.

Our results for the longitudinal spin dynamic structure factor are shown in Fig. 7, for both the symmetric ($k_y = 0$) and antisymmetric ($k_y = \pi$) channels, together with the linear spin-wave (SW) dispersion. We show a similar color density plot in Fig. 8 focusing on the antisymmetric sector with $k_y = \pi$. Notice that the SW results agree very well with the DMRG

data in the gapless phase, but as the gap open, the differences become more obvious, since spin-wave theory cannot describe the gapped phase of the Heisenberg ladder [9]. Elementary excitations on a two leg ladder are conventionally understood as rung triplons: a spin will pair with another one on the opposite leg forming a triplet excitation that costs an energy $\Delta \sim J$. The energy is lowered by propagating the triplet via spin-flips, in what can be qualitatively interpreted as a hard-core boson moving in a vacuum of rung-singlets. For $\alpha > \alpha_c$ we observe a gapped coherent band in the symmetric channel with vanishing spectral weight around $k_x = 0$, since $S_{total}^z = 0$. The antisymmetric channel presents coherent features at high energies, but the spectrum broadens as the momentum approaches $\vec{k} = (\pi, \pi)$ (this is more clearly seen in Fig. 7(g)).

As the value of $\alpha$ is reduced and approaches the quantum critical point, the two dispersive branches condense at $\vec{k} = (0,0)$ and $(\pi, \pi)$, respectively. The excitations display a sharp elastic peak at the ordering vector $(\pi, \pi)$, and we can observe how the bandwidth increases.

Interestingly, the width of the continuum in the symmetric $k_y = 0$ channel seems to get smaller as we approach $\vec{k} = (0, \pi)$ and both the magnon band and the spinon continuum seem to merge into a single sharp coherent dispersion. We also notice that the high energy features near the center of the Brillouin zone evolve adiabatically and are insensitive to the phase transition. It is thus reasonable to assume that in this region, magnons and triplons do not differ qualitatively. In fact, the same could be said about the symmetric branch, and the main distinction becomes question of semantics: in one case they are gapless, and in the other gapped, but otherwise, they are both interpreted as bound states of spinons.

In Fig. 8 we observe a very sharp peak at the ordering vector and a range of intermediate values of energies with little spectral weight below what looks like a separate branch. Since spin-wave theory is expected to work in the ordered phase, there is in principle no reason to expect two dispersive branches. Another, more reasonable possibility, is that in reality the space between the upper coherent band and the large elastic peak is occupied by an incoherent continuum with very small spectral weight, but finite size effects should not be discarded. Unfortunately, our limited resolution and the sublinear dispersion with a large slope near $\omega = 0$ prevent us from fully answering this issue.

## 5 Summary and Conclusions

Our numerical evidence points at a second order phase transition at $\alpha_c \sim 2.5$ from a gapped, magnetically disordered rung dimer phase with triplon excitations, to an antiferromagnetic phase with long range order and magnon excitations. However, the possibility of a weak first order transition should not be discarded. Our results in Fig. 1 are conspicuous enough to grant the question: is there a gap opening for $\alpha < \alpha_c$? If we trust that our extrapolation to the thermodynamic limit is indeed within error bars, this is definitely possible. In the quantum magnetism folklore, it is assumed that symmetry breaking is directly associated to the presence of gapless Goldstone modes [51–54]. However, it is easy to see that in the case of $\alpha = 0$ our model would realize symmetry breaking, but also that the energy would become superextensive, with a huge gap to the first excitation proportional to the system size [29]. The presence of a gap in systems with long range interactions should not come as a surprise; after all, Goldstone's theorem relies on the condition that the Hamiltonian is relatively local, with short range interactions (rigorously speaking, the soft modes should no longer be referred-to as "Goldstone modes" in the presence of non-local interactions). In addition, the assumption that the spin-wave dispersion should be linear is no longer valid in our case.

While triplons are intuitively easy to visualize as rung triplet excitations that propagate coherently, spin-waves are rather understood as fluctuations of the order parameter around

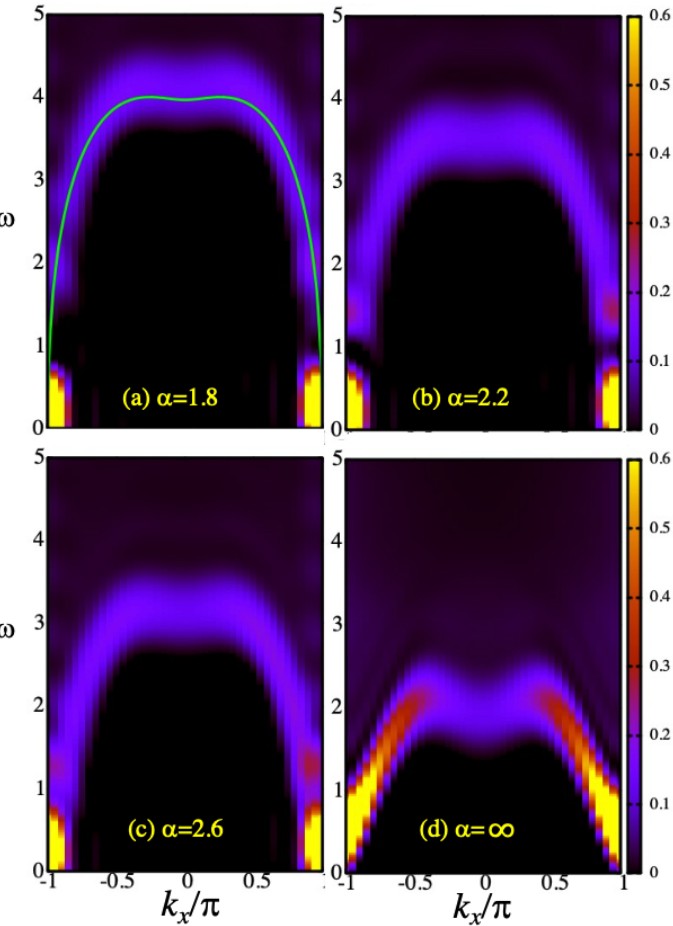

Figure 8: Longitudinal dynamic spin structure factor $S(k, \omega)$ with $k_y = \pi$, for a $20 \times 2$ ladder with long range interactions and different exponent $\alpha$ across the quantum critical point, obtained with tDMRG.

a symmetry broken ordered state [51, 52]. In the case of the pressure induced transition in TlCuCl$_3$, the triplon excitations condense at the transition and become gapless spin waves on the ordered phase [1] and excitations remain coherent throughout the transition. However, we notice that the "disordered" phase of our ladder system realizes hidden topological order characterized by a non-local string order parameter. Thus, one question that emerges from our studies is whether deconfined criticality can be realized or not [55–59]: while Landau's arguments forbid a direct second order phase transition between phases with order parameters that describe different symmetries, it is possible that in certain cases the transition could be continuous and that, when this occurs, quasiparticle excitations would not be well defined at the critical point, with the spectrum displaying a broad incoherent continuum. While these arguments rely on a direct transition between two ordered phases with incompatible local ordered parameters, in our case one of the phases has topological order. In our results, the peculiar features observed in the spectrum around $\vec{k} = (\pi, \pi)$ offers suggestive evidence of deconfined excitations, possibly in terms of spinons that carry spin $S = 1/2$ [60]. A deconfined critical point would also be characterized by a critical exponent $z = 1$, but our results are not conclusive. It is natural to ask whether the critical point can be identified with a conformal field theory or a new kind of criticality, but we do not have enough information to answer this question, since the algebra is not well defined in a finite volume because the theory is non-local. Quantum criticality connecting a topological ordered phase and a conventional

Landau ordered phase could represent a new paradigm in our understanding of quantum phase transitions.

In systems with long range interactions one typically finds sublinear dispersion with $z < 1$ [30, 61–69]. However, the anti-ferromagnetic transverse field Ising chain with long-range interactions shows critical exponents that correspond to the standard 1D transverse field Ising chain indicating the possibility of a CFT critical point in a long-range interacting model [70]. More work needs to be done in order to establish the universality class of the transition.

## Acknowledgments

LY and AEF acknowledge support from the National Science Foundation under grant No. DMR-1807814. The authors thank S. Sachdev, A. W. Sandvik, E. Katz, L. Manuel, and A. Trumper for useful discussions.

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
