# Peer review of "Topological to magnetically ordered quantum phase transition in antiferromagnetic spin ladders with long-range interactions"

_SciPost Physics, doi:SciPost Phys. 13, 060 (2022)_

## Round 2 · Referee Report · Anonymous · 2021-12-14

Strengths
1.- The paper is interesting and topical.
2.- The analysis is trustworthy and meticulous.
Weaknesses
No significant weaknesses.
Report
The physical system being studied is quite interesting. I am happy to recommend the paper for publication. The quality of the finite-size analysis is particularly good. The limitations of the conclusions drawn are clearly stated and the remaining open questions are discussed.
Requested changes
General grammar improvements only.
Author: Phillip Weinberg on 2022-04-21 [id 2405]
(in reply to Report 3 on 2022-03-21)We thank Referee 3 for carefully reading the manuscript and his/her constructive criticism. We hereby address his/her main concerns, hoping that it can now be accepted as is.
Referee; "the writing is poor and not very accessible..."
Authors: We agree that some parts of the manuscript require revisions, but we believe it should be perfectly accessible to a specialized audience. We already made considerable corrections in the new version, and re-written entire paragraphs, as detailed in the previous resubmission and we describe below. In addition, we point out that, to the best of our knowledge, this is the first time this model has been studied in the literature. It provides an unprecedented one-dimensional example of a transition from a gapped topological phase to one with spontaneous symmetry breaking, possibly of second order. We are very much convinced that our work may appeal to both broad, and specialized audiences and may provide the test ground for new field theories.
Ref: In the introduction, the authors should provide more details and background to make the paper more accessible to non-specialists. For instance, they should specify what "in a RPA-like sense" means; the meaning of "strong rung coupling limit"; and the definition of J_rung.
Authors: We thank the Referee for pointing out this omission. We re-wrote this paragraph and we hope that the new version clarifies this point:
"One could in principle envision such interactions emerging from a proximity coupling with a higher dimensional antiferromagnet or other ladders in a perturbative sense. The only free parameter in the problem is the exponent $\alpha$; for large $\alpha$ we expect the ground state to be in the same phase as the conventional Heisenberg ladder and the physics is well understood: the correlation length is short, of a few lattice spaces, and the gap is of the order of the coupling $J$ [9–20]. This ground state is adiabatically connected to the trivial limit of the conventional Heisenberg ladder corresponding to anisotropic couplings along the legs and rungs $J_{rung} \gg J_{leg}$. In this ''strong rung coupling limit'' the ground state is a product of rung dimers, the single-triplet gap is of order $\mathcal{O}(J_{rung})$ and excitations are rung triplets that can propagate coherently along the ladder." ... "On the other hand, in the model described by Eq.(1) we anticipate that the all-to-all unfrustrating interactions will yield a ground state with long-range AFM (N\'eel) order and gapless excitations for relatively small $\alpha$. These expectations are based on previous studies of Hamiltonian (1) in 1D chains [29–34], where a transition between a gapless spin-liquid and a gapless ordered phase was revealed. "
Notice that we have also added a figure (new Fig. 1) describing the geometry of the interactions.
Ref: On page 4, the paragraph starting with "In Fig. 1..." should be rewritten. It is the first paragraph presenting the numerical results, and it is written in a very obscure way.
Authors: We agree. This is the new version:
"In Fig. 2 we show both $B$ and $R$ as a function of $\alpha$ for various system sizes. We observe that $B$ monotonically increases for decreasing $\alpha$. This behavior indicates the onset of long-range N\'eel order for small values of $\alpha$. On the other hand, $R$, which is always negative by definition, is growing in absolute value as $\alpha$ decreases. For larger values of $\alpha$, in the gapped phase, $R$ tends to $0$ with increasing system size, implying that $O>0$. In the N\'eel phase (small $\alpha$) $R$ converges to a finite value, indicating that $O=0$. The ''steepness'' of the $B$ and $R$ curves increases with increasing system size. This observation is consistent with the finite-size behavior one would expect from a phase transition [43]. "
Ref: On page 6, in the first paragraph of part 3, the statement that the ground state entanglement entropy obeys a volume law due to long-range interactions is stunning. This would be an extraordinary observation. I suspect it is not true, and the entanglement entropy obeys in fact a log(L) scaling, as found in all sudies of long-range models. This should be clarified.
Authors: We point out that we never claimed to see a volume law. We believe that we never said anything incorrect. What we implied is that, in the presence of long range interactions one may naively expect a volume law behavior. In fact, we observe that "Surprisingly, the entanglement entropy does not grow dramatically in the gapless phase and across the transition." We have rephrased the text in a way that may sound less confusing:
"What can make this problem particularly challenging is the possibility of a volume law entanglement law due to the presence of all-to-all interactions. However, in the gapped phase, the correlation length remains finite and the entanglement remains under control. Surprisingly, the entanglement entropy does not grow dramatically in the gapless phase and across the transition. This may appear to be a general feature of one-dimensional models with long-range interactions as has been observed in quantum spin chains, which display a $log(L)$ behavior [33, 44–46]."
We added these three references that we considered relevant. Notice that the only rigorous results for the scaling of the entanglement entropy in systems with long range interactions is presented in the new reference Gong2017, which is valid for exponent alpha > 2. The other results are empirical and come from numerical simulations, some out of equilibrium, and some in ferromagnetic models.
Ref: On the same page 6. What polynomial fit is done for the gap to extrapolate finite-size data? In Fig.4 (inset), it seems that the gap is linear in 1/L. But this contradicts the scaling 1/Lz, which would imply a vertical slope in 0. This should be clarified, and possibly corrected.
Authors: We clarified that we used a second order polynomial (this information was in the figure caption but not the body of the manuscript).
Concerning the scaling, we emphasize that in reality the behavior should be 1/L^z with z<1. This is pretty obvious in the spectral function and is a general feature of one-dimensional systems with long-range interactions. The apparent linear behavior for large L is an artifact of this extrapolation. As a matter of fact, our extrapolation of z in the (newly renumbered) Figure 6 clearly shows both facts: z <1, and a power law extrapolation is ill behaved due to the small dataset. Similarly, a fit using Eq.(8) is ill behaved and does not yield physical results.
In short: a polynomial fit is not accurate and cannot describe the proper scaling for large L, where the sub-linear corrections become dominant. Consider a f(x)=sqrt(x), for instance: if one has data for x > 5, one would never see the sub-linear behavior and would completely miss the correct scaling near the origin. In that case, it is tempting to carry our a quadratic fit. In our case we are subject to the same dilemma: we cannot observe the sub-linear scaling in the dataset because we cannot simulate sufficiently large ladders. That's why the extrapolation of the gap should be taken with a grain of salt and is not very accurate. That is also why we had to proceed to other strategies in order to properly account for these corrections, as we discuss in great detail in the paragraph starting right above Eq.(8).
Ref: Why is the extrapolation of z(L) to L→∞ not showed? A plot should be done to illustrate this extrapolation and therby complement Fig. 4 (right panel)
Authors: We answered this in the previous point. In short: it is pointless because the extrapolation to the thermodynamic limit is not possible due to the huge variability based on the type of extrapolation used. In the newly labeled Fig. 6 we demonstrate this issue explicitly by showing that the power-law and polynomial extrapolations give wildly different results when extrapolating the dynamic exponent. On the other hand, the critical point value seems much less sensitive to the extrapolation method used and both give consistent results that are within error bars.
Summary of changes:

---

## Round 2 · Referee Report · Anonymous · 2022-2-6

Report
The authors consider a 2 leg ladder of spins with long-range Heisenberg interactions that decay with the distance $r$ with a power law $1/r^\alpha$. They use quantum Monte Carlo and DMRG for computing the phase diagram as a function of the exponent $\alpha$, the dynamic structure factor as well as the nature of the excitations. They find a quantum phase transition between a Neel and a "rung-dimer" phase when the exponent $\alpha$ reaches the value of $\sim 2.5$. The phases are detected by looking at the behaviour of the staggered magnetization (for the Néel phase) and a string order parameter (for the "rung-dimer").
Although some of the results are a generalization of earlier work, I believe that this work can be of potential interest for a broad audience. Before recommending publication, I would the authors clarify the following point.
The main concern I have is on the nature of the transition between the two phases. The authors claim it is a second order phase transition by computing the Binder cumulant $B$ and an indicator $R$ related to the string order parameter. They then locate the transition in $\alpha$ by looking at the point where the curves which represent B and R cross for different system sizes. By looking at Fig. 1 while this crossing is evident for $R$, it is not clear that it happens also for $B$. Moreover, the ground state gap (Fig. 4) does not seem to close in the vicinity of the transition. I think the authors should exclude the possibility that the change of the phase when $\alpha$ decreases is a simple crossover and not a phase transition.
I have also few remarks
1. There's a space missing in the abstract (6th line)
2. I would add a figure in order to explain the interactions of Hamiltonian (1).
3. page 4, line 4 affect -> effect
4. page 4, paragraph Considering a functional... The symbol $O_l$ is repeated twice
5. In the captions of fig 2 and 3 the symbol chi is not typed correctly.
6. Page 6, paragraph As discussed..., "the limit alpha" -> "in the limit alpha"
Author: Phillip Weinberg on 2022-03-08 [id 2273]
(in reply to Report 2 on 2022-02-06)
We thank the referees for their valuable comments and, in particular, for recommending our manuscript for publication. We have addressed their concerns in our new version and in the following we proceed to answer Report 2 in detail:
REFEREE:
The main concern I have is on the nature of the transition between the two phases. The authors claim it is a second order phase transition by computing the Binder cumulant B and an indicator R related to the string order parameter. They then locate the transition in α by looking at the point where the curves which represent B and R cross for different system sizes. By looking at Fig. 1 while this crossing is evident for R, it is not clear that it happens also for B.
RESPONSE:
While we agree that in Fig. 1 it is a bit hard to see the crossing points, results for B and R from our large scale QMC calculations are shown in more detail in Fig. 2. These crossing points are obtained from a much more refined set of data compared to the points presented in Fig. 1.
REFEREE:
Moreover, the ground state gap (Fig. 4) does not seem to close in the vicinity of the transition. I think the authors should exclude the possibility that the change of the phase when α decreases is a simple crossover and not a phase transition.
RESPONSE:
We already discuss this issue in the manuscript, namely that finite-size corrections to the gap are strongly sub-linear, making an accurate extrapolation of the gap very problematic. A way to see this is by supposing that the scaling at large L (small 1/L) goes as sqrt(1/L). If L is not large enough, we would not be able to resolve the sublinear behavior, which is precisely the problem that affects our extrapolation here. We try to get around this issue by estimating the dynamic exponent. Similar to the plots for $B$ and $R$, the extrapolation of the "finite length" dynamic exponent shows crossing points such that on one side of the transition the extrapolation tends to a constant and on the other side it is tending to 0. This behavior indicates a transition from a gapped phase to a gapless phase with a finite staggered magnetization. Using the crossing points we can estimate the position of the critical point, which agrees with the QMC results for the order parameter. The issue of this critical point being of the Landau type, or a deconfined critical point, is left open as a possibility that deserves further study.
REFEREE:
I have also few remarks
- There's a space missing in the abstract (6th line)
- I would add a figure in order to explain the interactions of Hamiltonian (1).
- page 4, line 4 affect -> effect
- page 4, paragraph Considering a functional... The symbol Ol is repeated twice
- In the captions of fig 2 and 3 the symbol chi is not typed correctly.
- Page 6, paragraph As discussed..., "the limit alpha" -> "in the limit alpha"
RESPONSE:
We have updated the manuscript to fix these issues, including a new figure. The changes are reflected in arXiv v3.

---

## Round 2 · Referee Report · Anonymous · 2022-3-21

Strengths
1. The topic is interesting and the studied model is simple and relevant.
2. The quality of the numerics is good.
3. The limitations of the results are accurately discussed.
Weaknesses
1. The writing can be improved.
2. The introduction and presentation of the context is hardly accessible to a non-specialist.
3. Some statements are puzzling (and are in fact probably wrong).
Report
The paper is overall interesting, and the numerics is of good quality, but the writing is poor and not very accessible. After corrections are taken into accound, I believe tha paper is more suitable to Scipost Physics Core than to Scipost Physics.
1. In the introduction, the authors should provide more details and background to make the paper more accessible to non-specialists. For instance, they should specify what "in a RPA-like sense" means; the meaning of "strong rung coupling limit"; and the definition of $J_{rung}$.
2. On page 4, the paragraph starting with "In Fig. 1..." should be rewritten. It is the first paragraph presenting the numerical results, and it is written in a very obscure way.
3. On page 6, in the first paragraph of part 3, the statement that the ground state entanglement entropy obeys a volume law due to long-range interactions is stunning. This would be an extraordinary observation. I suspect it is not true, and the entanglement entropy obeys in fact a log(L) scaling, as found in all sudies of long-range models. This should be clarified.
4. On the same page 6. What polynomial fit is done for the gap to extrapolate finite-size data? In Fig.4 (inset), it seems that the gap is linear in $1/L$. But this contradicts the scaling $1/L^z$, which would imply a vertical slope in 0. This should be clarified, and possibly corrected.
5. Why is the extrapolation of $z(L)$ to $L\to \infty$ not showed? A plot should be done to illustrate this extrapolation and therby complement Fig. 4 (right panel).
Typos:
p. 2: "this behavior has has been experimentally observed"
p. 3: "the critical point using and the correlation length"
p. 4: "considering a functional form $O_l$ $O_l$
caption of Fig. 2 and 3: $chi^2 \to \chi^2$.
p. 5: "to generate the the mean"

---

## Round 4 · Referee Report · Anonymous · 2022-6-24

Report

The new version of the paper is very clear, and the authors have taken into account the remarks made by the various Referees. I recommend publication.

---

## Round 4 · Referee Report · Anonymous · 2022-7-4

Report

The authors did a great work for improving the readability of the manuscript and they have clarified several issues with further explanations. I recommend the manuscript for publication in its current version. I would however agree with Referee 3 that this paper is more suitable to Scipost Physics Core, by looking at its acceptance criteria.

---

## Round 4 · List of Changes

List of changes by section between v1 and v4, note that the references numbering have changed.

###################################################
###################################################

Introduction:

###################################################
###################################################

- "Ramarkably" -> "Remarkably"
- "algebraically decaying couplings:" -> "additional algebraically decaying all-to-all couplings:"
- added new figure 1 and additional text: "(See Fig. 1 for a graphical representation)"
- "RPA-like" -> "perturbative"
- the sentences:

"In the strong rung coupling limit, the ground state is a product of rung dimers, and excitations are rung triplets that can propagate coherently along the ladder. On the other hand, when the chains are decoupled, excitations are deconfined spinons that carry spin $S=1/2$. "

is replaced by:

"This ground state is adiabatically connected to the trivial limit of the conventional Heisenberg ladder corresponding to anisotropic couplings along the legs and rungs $J_{rung} \gg J_{leg}$. In this ``strong rung coupling limit'' the ground state is a product of rung dimers, the single-triplet gap is of order $\mathcal{O}(J_{rung})$ and excitations are rung triplets that can propagate coherently along the ladder."

- "were noticed while back" -> "has been established for some time now"

- the sentences:

"On the other hand, for relatively small α, we anticipate a ground state with long-range AFM (Néel) order and gapless Goldstone modes. These expectations are based on previous studies of Hamiltonian (1) in 1D chains [29–34], where a transition between a gapless spin-liquid and a gapless rdered phase was revealed."

is replaced by:

"On the other hand, in the model described by Eq.(1) we anticipate that the all-to-all unfrustrating interactions will yield a ground state with long-range AFM (Néel) order and gapless excitations for relatively small α. These expectations are based on previous studies of Hamiltonian (1) in 1D chains [29–34], where a transition between a gapless spin-liquid and a gapless ordered phase was revealed."

- "study properties of" -> "understand"
- "is studied" -> "is discussed"

###################################################
###################################################

Quantum critical point:

###################################################
###################################################

- first paragraph changes:

original paragraph:

In this section, we present several complementary methods to estimate the position of the
quantum critical point $\alpha_c$. We first focus on determining the critical point using and the
correlation length exponent ν using QMC on periodic ladders of length up to $L = 96$.
We also develop a method of determining the critical point using the dynamic exponent
calculated from finite size gaps obtained from DMRG [39, 40] calculations.

The new paragraph:

In this section, we study the transition between the Néel phase and the rung-dimer phase.
In order to exclude the possibility of an intermediate phase, we use both order parameters
to calculate the critical point $\alpha_c$ and the correlation length exponent $\nu$. We shall show that
both order parameters give the same $\alpha_c$ and $\nu$ providing evidence of a direct order-to-order
transition.

- second paragraph changes:

original paragraph:

To study the ground state properties we use projector QMC with a trial state that is given by an amplitude product state in the valance bond basis[31,41]. We use the same trial state for all values of $\alpha$. This state has long range N\'eel order to help reduce the number of projector steps needed to reach the ground state in the N\'eel phase. In the rung-dimer phase, this trial state has little affect on the number of projector steps required because of the finite gap in the thermodynamic limit. We study two different order parameters that define the order on either side of the transition. If the transition point is determined to be the same using both order parameters, we can exclude the possibility of an intermediate phase between the rung-dimer and N\'eel phases. For the N\'eel order we use the Binder Cumulant defined as:
\begin{equation}
B = \frac{3}{2}\left(1-\frac{1}{3}\frac{\langle M_s^2\rangle}{\langle\vert M_s\vert\rangle^2}\right),
\end{equation}
where $M_s=\sum_{i}(-1)^{x_i+y_i}S^z_i$. In our QMC simulations we
%use the improved estimators when calculating $\langle|M_s|\rangle$ and $\langle M_s^2\rangle$ exploiting
exploit the full $SU(2)$ symmetry of the ground state~[31,41].

The new paragraph:

To perform these calculations, we use standard finite-size scaling (FSS) methods[39]. We calculate the quantities of interest using projector QMC applied to Eq.(\ref{rkky}) with periodic boundary conditions\footnote{We enforce the boundary conditions in the definition of the distance between the sites, $\vert\vec{r}_i-\vec{r}_j\vert$, being the minimum of the two possible distances.}. We start the projector QMC with a initial state expressed as an amplitude product state in the valance bond basis[32,40,41]. We use the same initial state for all values of $\alpha$. We choose the amplitudes such that the initial state has long-range N\'eel order to reduce the number of projector steps needed to reach the ground state in the gapless N\'eel phase. The gap opens up in the rung-dimer phase; as such, the initial state has little effect on the number of projector steps required to reach the ground state.

To calculate the critical point using the N\'eel order parameter we use the Binder Cumulant defined as:
\begin{equation}
B = \frac{3}{2}\left(1-\frac{1}{3}\frac{\langle M_s^2\rangle}{\langle\vert M_s\vert\rangle^2}\right),
\end{equation}
where $M_s=\sum_{i}(-1)^{x_i+y_i}S^z_i$. In our QMC simulations we
%use the improved estimators when calculating $\langle|M_s|\rangle$ and $\langle M_s^2\rangle$ exploiting
exploit the full $SU(2)$ symmetry of the ground state[40,41].

- third paragraph changes:

original paragraph:

For the rung-dimer phase we use the string order parameter $O$, defined in Eq.(3). Much like a correlation function, we observe that in a finite size system $O_l$ has a non-vanishing value due to finite-size effects. To systematically study the convergence to the thermodynamic we look at the ratio between $O_l$ measured at two lengths, $L/2$ and $L/4$, and analyze the following quantity as a function of $\alpha$ and system size $L$[41]:
\begin{equation}
R = \log\left(\frac{O_{L/2}}{O_{L/4}}\right).
\end{equation}
In order to measure $O_l$ we use the standard estimator calculated from the $S^z$ basis of the QMC simulation. Because of the non-local nature of this order parameter, the results have more noise compared to the Binder cumulant.

new paragraph:

On the other hand, the rung-dimer phase characterized by a string order parameter $O$, defined in Eq.(3). Much like a correlation function, we observe that in a finite size system with finite $l$, $O_l$ has a non-vanishing value even in the N\'eel phase. We suspect this is due to finite-size effects. In the same spirit as the correlation length discussed in Ref. [39] we study the log-ratio between $O_l$ measured at two values of $l$:
\begin{equation}
R = \log\left(\frac{O_{L/2}}{O_{L/4}}\right).
\end{equation}
In order to calculate $O_l$ in our QMC simulations we use the standard estimator calculated in the $S^z$ basis. Because of the non-local nature of this order parameter, the results have more noise compared to $B$.

- fourth paragraph changes:

original paragraph:

Considering a functional form $O_l$ $O_l=f_l+C$ (where $f_l$ is an asymptotically decaying function) there are a few possible outcomes for $R$ in the thermodynamic limit. For $C>0$, $R\rightarrow 0$ as $L\rightarrow\infty$. In the other hand, for $C=0$ the asymptotic behavior of $R$ is determined by the behavior of $f_l$. If $f_l$ decays as a stretched exponential (or faster) $R$ diverges as $L\rightarrow\infty$, while for $f_l$ decaying slower, $R\rightarrow {\rm const}\le 0$, {\it i.e.} for a power-law decay this constant will be negative.

new paragraph:

Before we show the results from our simulations, we discuss how to interpret $R$ as a function of system size $L$. There are few possible outcomes for $R$ in the thermodynamic limit depending on the asymptotic behavior of $O_l$. If $O_l$ tends to a constant as $l \rightarrow \infty$, by definition $R \rightarrow 0$. Otherwise, because of the $\log$, if $O_l$ decays exponentially or faster, then $R$ diverges as $L\rightarrow\infty$. If $O_l$ decays slower than exponential, $R\rightarrow {\rm const}\le 0$, {\it i.e.} for a power-law decay this constant will be negative.

- fifth paragraph changes:

original paragraph:

In Fig. 1 we show both $B$ and $R$ as a function of $\alpha$ for various system sizes. We see that as $\alpha$ increases, $B$ monotonically increases as alpha decreases below $\alpha \sim 2.5$, indicating the onset of long-range N\'eel order for lower values of $\alpha$. On the other hand, $R < 0 $, growing in absolute value $\vert R\vert$ as $\alpha$ increases. Another thing to note is that the value of $R$ seems to be converging to a finite value below $\alpha=2.5$ indicating that the string order parameter decays as a power-law in this regime. For both $B$ and $R$ the crossover becomes sharper for larger system sizes which is an indication of a phase transition[42] from a magnetically ordered phase to a magnetically disordered phase. According to the behavior of $R$, this transition is from a phase with $O_l$ converging to a constant value at large $\alpha$ in the limit $l\rightarrow\infty$, to another one with $O_l$ decaying to $0$ as a power-law in $l$, implying that $O=0$ in the N\'eel phase.

new paragraph:

In Fig. 2 we show both $B$ and $R$ as a function of $\alpha$ for various system sizes. We observe that $B$ monotonically increases for decreasing $\alpha$. This behavior indicates the onset of long-range N\'eel order for small values of $\alpha$. On the other hand, $R$, which is always negative by definition, is growing in absolute value as $\alpha$ decreases. For larger values of $\alpha$, in the gapped phase, $R$ tends to $0$ with increasing system size, implying that $O>0$. In the N\'eel phase (small $\alpha$) $R$ converges to a finite value, indicating that $O=0$. The ``steepness'' of the $B$ and $R$ curves increases with increasing system size. This observation is consistent with the finite-size behavior one would expect from a phase transition[39].

- removed the sixth paragraph:

Both $B$ and $R$ represent different types of order and they serve as means to independently determine the critical point. If this transition is between two ordered phases one would expect that $B$ and $R$ will indicate the same critical point and correlation length exponent $\nu$. %In order for this transition to be of second order one would expect that not only the critical points for the two order parameters are the same, but also the correlation length exponent $\nu$.
We have specifically chosen the forms for $B$ and $R$ to allow us to systematically extract the critical point and correlation length exponent from finite-size calculations.

- seventh paragraph changes:

original paragraph:

To determine the location of the critical point we look at the points at which two curves (either $B$ or $R$) corresponding to different system sizes cross. Because of finite-size scaling (FSS) corrections the crossing points will drift closer to the critical point as the system sizes increase~\cite{campostrini14}. Specifically, we look at the crossing points between curves corresponding to system sizes $L$ and $2L$. We will denote this crossing point as $\alpha^*(L)$.

new paragraph:

To determine the location of the critical point, we look at the crossing points between two curves (either $B$ or $R$) corresponding to different system sizes. Because of FSS corrections, the crossing points will drift closer to the critical point as the system sizes increase~\cite{campostrini14}. Specifically, we look at the crossing points between curves corresponding to system sizes $L$ and $2L$, which we denote as $\alpha^*(L)$.

- paragraphs eight and nine changes:

original paragraphs:

We can also extract the correlation length exponent $\nu$ from $B$ and $R$ because they both have a scaling dimension of $0$. From the FSS form for this type of observable it is possible to show that the value of $\nu$ can be determined by the following limit:
\begin{gather}
\nu = \lim_{L\rightarrow\infty} \nu^*(L) \\ \nu^*(L) = \left[\log_2\left(\frac{\partial_\alpha Y(\alpha^*(L),2L)}{\partial_\alpha Y(\alpha^*(L),L)}\right)\right]^{-1}\label{eq:nu_estimate}
\end{gather}
where $Y(\alpha,L)$ corresponds to either $B$ or $R$. This can easily be seen by looking at the FSS form for an observable with scaling dimension $0$[42].

Practically speaking we can only study a finite number of values of $\alpha$, so we interpolate those points with a polynomial. Using the interpolation for $L$ and $2L$ we can calculate both $\alpha^*(L)$ and $\nu^*(L)$. To account for the statistical errors coming from the QMC sampling we use the bootstrapping method. This involves drawing a new set of values for $B$ and $R$ from a normal distribution with a mean and standard deviation given by the QMC mean and standard error for each point respectively. After drawing the new points, a polynomial is fitted from which $\alpha^*(L)$ and $\nu^*(L)$ are obtained. This procedure is repeated for many random realizations of the data points. Each realization has independent values of $\alpha^*(L)$ and $\nu^*(L)$. From this set of values the mean and standard deviation are calculated. In this work we use $10000$ realizations to generate the the mean and standard deviation corresponding to the points and error bars shown in Figs. 2 and 3.

new paragraph:

By construction, both $B$ and $R$ have no multiplicative scaling factor as a function of $L$ in their respective FSS form. As such, it is possible to use them to extract the correlation length exponent, $\nu$, from the following limit:
\begin{gather}
\nu = \lim_{L\rightarrow\infty} \nu^*(L) \\ \nu^*(L) = \left[\log_2\left(\frac{\partial_\alpha Y(\alpha^*(L),2L)}{\partial_\alpha Y(\alpha^*(L),L)}\right)\right]^{-1}\label{eq:nu_estimate}
\end{gather}
where $Y(\alpha,L)$ corresponds to either $B$ or $R$~\cite{campostrini14}. In practice one can only study a finite number of values of $\alpha$, we interpolate those points with a polynomial. Using the interpolation for $L$ and $2L$ we can calculate both $\alpha^*(L)$ and $\nu^*(L)$. To account for the statistical errors coming from the QMC calculations we use the bootstrapping method. This involves drawing a new set of values for $B$ and $R$ from a normal distribution with a mean and standard deviation given by the QMC mean and standard error for each point respectively. After drawing the new points, a polynomial is fitted from which $\alpha^*(L)$ and $\nu^*(L)$ are obtained. This procedure is repeated for many realizations of the data points. From this set of values the mean and standard deviation are calculated. In this work we use $10000$ realizations to generate the mean and standard deviation corresponding to the points and error bars shown in Figs. 3 and 4.

- tenth paragraph changes:

original paragraph:

The results for $\alpha^*(L)$ and their respective estimates in the thermodynamic limit for both $B$ and $R$ are show in Fig. 2. The extrapolation is done using a power-law fit of the form $\alpha^*(L) = \alpha_c + b/L^\gamma$ for both, $B$ and $R$. The two order parameters give a critical point of $2.520$ within error bars. We also use a linear expression to extrapolate the values of $\nu^*(L)$ for both $B$ and $R$, as shown in Fig. 3, yielding a value of $\nu=1.79$ within error bars. These results are a strong indication that there is a direct transition between the rung-dimer phase and the N\'eel phase. Our next goal is to establish if the transition is continuous or first order.

new paragraph:

In Fig. 3 we show $\alpha^*(L)$ for both $B$ and $R$ as a function of $1/L$. The curves going through the points depict a power-law fit ($\alpha^*(L) = \alpha_c + b/L^\gamma$) of the finite-size data extrapolated to the thermodynamic limit. We obtain a critical point of $2.520$ (within error bars) for both order parameters. Turning to the correlation exponent $\nu$, in Fig. 4 we show $\nu^*(L)$ versus $1/L$ for both $B$ and $R$. The lines in the figure depict the linear fit of the finite-size data extrapolated to the thermodynamic limit. Much like the critical point, both order parameters give a value of $\nu=1.79$. Our numerical results show that both order parameters point to the same critical point with the same correlation length exponent providing strong evidence of a direct transition between the rung-dimer phase and the N\'eel phase. Our next goal is to establish if the transition is continuous or first order.

###################################################
###################################################

Gap and dynamic exponent $z$:

###################################################
###################################################

- "...is another quantity that..." is removed
- "What makes..." -> "What can make..."
- "...for DMRG is the volume..." -> "...is the possibility of a volume law..."
- "..., as also observed on the 1D case" is removed
- new sentence: "This may appear to be a general feature of one-dimensional models with long-range interactions as has been observed in quantum spin chains, which display a log(L) behavior [33, 44–46]."
- removed "...and the gap as a function of $1/L$ for various $\alpha$ in the inset"
- "To carry out the extrapolation to the thermodynamic limit we use a a polynomial fit of finite-size data as a function of $1/L$." -> "To carry out the extrapolation to the thermodynamic limit we use a second order polynomial fit of finite-size data as a function of $1/L$, as shown in the inset."
- "We do not see the gap closing at $\alpha = 2.520$ due to the strong sublinear scaling of the gap." -> "We do not see a closing of the gap at $\alpha = 2.520$ due to the strong sublinear scaling behavior that introduces corrections that require larger system sizes (as we describe below, a power-law extrapolation is ill-conditioned for this small dataset)."
- "Given the power-law nature of the interactions, the finite-size effects are much stronger compared to a local Hamiltonian making the dynamic exponent difficult to extract" -> "Given the power-law nature of the interactions, the finite-size effects are much stronger compared to a local Hamiltonian making the dynamic exponent difficult to extract when fitting the gap directly."
- "As
a result, we use the crossing points between two system sizes to extrapolate the location of the critical point." -> "As a result, the crossing points between any two finite-size curves will converge to the exact critical point in the limit where both system sizes go to infinity, giving us another independent method to calculate the location of the critical point."
- "...crossing points..." -> "...behavior of the finite-size curves..."
- "To fit the data we use an expression $y=y_0+b/L^\beta$ due to the limited number of points. We find that the critical point is around $2.5$ based on the two different extrapolation methods." -> "To fit the finite-size data we use a power-law ({\it e.g.} $y=y_0+b/L^\beta$), and a quadratic fit. We find that the critical point is around $2.5$ based on the two different extrapolation methods. "
- "...apply the extrapolation to..." -> "...extrapolate the..."
- "...like..." -> "...as we did for..."

---

## Editorial Decision

published